# The Role of Reactive Oxygen Species in Age-Related Macular Degeneration: A Comprehensive Review of Antioxidant Therapies

**DOI:** 10.3390/biomedicines12071579

**Published:** 2024-07-16

**Authors:** Merve Kulbay, Kevin Y. Wu, Gurleen K. Nirwal, Paul Bélanger, Simon D. Tran

**Affiliations:** 1Department of Ophthalmology & Visual Sciences, McGill University, Montréal, QC H4A 3S5, Canada; merve.kulbay@mail.mcgill.ca; 2Division of Ophthalmology, Department of Surgery, University of Sherbrooke, Sherbrooke, QC J1H 5N4, Canada; yang.wu@usherbrooke.ca (K.Y.W.);; 3Department of Zoology, University of British Columbia, Vancouver, BC V6T 1Z4, Canada; 4Faculty of Dental Medicine and Oral Health Sciences, McGill University, Montreal, QC H3A 1G1, Canada

**Keywords:** age-related macular degeneration, oxidative stress, antioxidants, treatments

## Abstract

This review article delves into the intricate roles of reactive oxygen species (ROS) in the pathogenesis of age-related macular degeneration (AMD). It presents a detailed analysis of the oxidative stress mechanisms that contribute to the development and progression of these diseases. The review systematically explores the dual nature of ROS in ocular physiology and pathology, underscoring their essential roles in cellular signaling and detrimental effects when in excess. In the context of AMD, the focus is on the oxidative impairment in the retinal pigment epithelium and Bruch’s membrane, culminating in the deterioration of macular health. Central to this review is the evaluation of various antioxidant strategies in the prevention and management of AMD. It encompasses a wide spectrum of antioxidants, ranging from dietary nutrients like vitamins C and E, lutein, and zeaxanthin to pharmacological agents with antioxidative properties. The review also addresses novel therapeutic approaches, including gene therapy and nanotechnology-based delivery systems, aiming to enhance antioxidant defense mechanisms in ocular tissues. The article concludes by synthesizing current research findings, clinical trial data, and meta-analyses to provide evidence-based recommendations. It underscores the need for further research to optimize antioxidant therapies, considering individual patient factors and disease stages. This comprehensive review thus serves as a valuable resource for clinicians, researchers, and healthcare professionals in ophthalmology, offering insights into the potential of antioxidants in mitigating the burden of AMD.

## 1. Introduction

The posterior eye segment encompasses various ocular pathologies, with age-related macular degeneration (AMD) being the most common. AMD is amongst the leading causes of blindness globally [1]. AMD progression is highly dependent on oxidative stress. AMD refers to macular degeneration with or without neovascularization. As the disease progresses, it prevents the passage of light, therefore leading to blindness if left untreated. The burden associated with this disease cannot go unnoticed; the disability-adjusted life years (DALY), which represents the sum of the years of life lost due to premature mortality and the years lived with a disability (World Health Organization), was shown to have increased for AMD [2]. Although numerous efforts have been deployed in the past decades to increase AMD treatment efficiency and outcomes, a significant disparity in AMD therapies exists based on socioeconomic status and racial inequities [3,4,5].

Multiple risk factors are accountable for the development of AMD, the most well-established cause being age-related modifications [6]. Other risk factors where an association was shown over the years include genetic predispositions [7], dietary intakes (e.g., a high Mediterranean diet decreases the risk of progression of AMD) [8], smoking [9,10], alcohol [11,12], and the presence of cardiovascular disease [13]. However, the association between AMD and sunlight exposure [6], the presence of cardiovascular disease [14,15], alcohol [11,16], or diet [6] remains controversial.

A common landmark to these etiologies is the formation of reactive oxygen species (ROS). Numerous studies have highlighted the importance and center role of ROS in AMD [17,18]. Factors disrupting the redox balance, such as the depletion of antioxidants and subsequent accumulation of ROS, contribute to disease progression (Figure 1). Disruptions in the cellular signaling pathways involved in cell proliferation and apoptosis, therefore leading to inflammation, are the contributory factors in the pathogenesis of ocular diseases. In this comprehensive literature review, we will review the pathogenesis of AMD, with an emphasis on the crucial role of ROS, discuss the current and novel antioxidant strategies for the prevention and management of this pathology with evidence-based recommendations, and finally, discuss the future perspectives of the clinical management of AMD.

## 2. Physiology of the Retina and Its Redox Regulatory Mechanisms

### 2.1. Structure and Function of the Retina

The posterior eye segment compromises most of the eye and encompasses the vitreous humor, the retina, the optic nerve, and the choroid. Vision can be divided into two categories—peripheral and central vision—which are both mediated in part by the retina. The retina, with a thickness of approximately 0.5 mm, is composed of 10 distinctive layers encompassing interconnected neurons by synapses, such as photoreceptors (i.e., cones and rods), and is the key player in the ability to see (Figure 2).

Phototransduction (e.g., conversion of light to a chemical signal) is mediated by cones and rods [19]. However, differences in light sensitivity in cones and rods exist; cones are less sensitive to light, therefore producing a milder and less efficient phototransduction cascade [19]. Structurally, rods and cones are highly similar (Figure 3). They are composed of a ciliary body, located in the outer nuclear layer (ONL), which contains the cell nucleus, and establish synapses with the bipolar cells in the outer plexiform layer (OPL) [20]. Orchestration of the phototransduction cascade occurs in the outer segment (OS) of photoreceptors [20]. On a histological scale, the OSs of photoreceptors are considered to be a modified sensory cilium [20]. The OS contains essential proteins and enzymes involved in the phototransduction cascade, such as opsins, phosphoinositide-3-kinase (PI3K), and protein kinase B (AKT) signaling components [21]. Nonetheless, the inner segment (IS) of photoreceptors is crucial, given that it contains the machinery for protein synthesis (e.g., endoplasmic reticulum, apparatus of golgi, and mitochondria) and is in dynamic communication with the OS of photoreceptors [20]. The viability and structural function of photoreceptors are dependent on the retinal pigment epithelium (RPE) layer and underlying vasculature system, the choroid [22]. The transport of oxygen, nutrients, and ions from the choroid to the photoreceptors is mediated through the RPE. The blood–retina barrier (BRB) contributes to the robust antioxidative system of the retina by transporting vitamins (e.g., vitamin E and ascorbate) and antioxidant enzymes (e.g., catalase, glutathione (GSH), glutathione peroxidase (GPx), and glutathione-transferases) [23].

As the human eye adapts to night- and day-vision, as well to peripheral and central visions, the regional distribution of cones and rods varies according to task-specificity [24]. The central part of the retina, also known as the macula, is involved in central vision. Pathologies disrupting the macular region, such as AMD, lead to central vision loss.

### 2.2. Metabolic Requirements and Regulation

Retinal metabolism is a highly regulated pathway that involves distinct key players. The RPE is the backbone of photoreceptor cell viability; glucose transport from the choroid to the photoreceptors through the BRB was shown to support photoreceptor cell viability [25,26,27]. The transport of glucose molecules across the blood–retina barrier was shown to be mediated by the glucose transporter 1 (GLUT1) receptor [28]. Glucose metabolism in the retina occurs through the aerobic glycolysis pathway [29]. Aerobic glycolysis, also known as the Warburg effect, is promoted in biological systems with a rapid cell turnover, such as the retina. Longoni et al. have reviewed the importance of glucose metabolism in retinal cells; photoreceptors require great levels of glucose to produce substrates for lipid synthesis, given the high activity level in the OSs of cones and rods [30]. Glycolysis end products, in particular lactate and pyruvate, were shown to promote oxidative stress resistance through the unfolded protein response (UPR) and subsequent activation of nuclear factor erythroid 2-related factor 2 (NRF2) [31]. Rods play a crucial role in the regulation metabolic metabolism by secreting modified thioredoxin, which is a rod-derived cone viability factor [32]. Modified thioredoxin is known to protect from ROS [32].

## 3. Ocular Damages Induced by Reactive Oxygen Species

The pivotal role of oxidative stress in the pathogenesis of numerous ocular pathologies has been thoroughly discussed and reviewed [33,34,35]. A well-known mechanism involved in AMD pathogenesis is the production of ROS, through oxidative stress-producing risk factors (i.e., smoking and metabolic syndromes) and the maintenance of a pro-inflammatory microenvironment within the retina through their positive feedback loop [36]. In this section, we review the main hallmarks of ROS-mediated AMD pathogenesis, with a focus on the most recent advances.

### 3.1. Pathogenesis of Age-Related Macular Degeneration

AMD can be divided into two subtypes: dry and wet AMD, also known as neovascular AMD (nAMD). The presence of neovascularization in wet AMD is the key differentiating factor between both subtypes. Additional key features of AMD involve the presence of retinal deposits, known as drusens, and geographic atrophy sparing or not sparing the macular region (Figure 4) [37].

The first step in the pathogenesis of AMD involves the deposition of drusen at the level of the retinal pigmented epithelium (RPE). Drusen consists of extracellular deposits encompassing proteins, cholesterol, apolipoproteins, and carbohydrates [38]. It is believed that drusen result from a disrupted lipid metabolism pathway due to aging and a pro-oxidative microenvironment. The pro-oxidative microenvironment due to chronic inflammation has been shown to positively regulate pro-inflammatory factor secretion from endothelial cells [39]. Retinal human tissue is rich in lipids; lipid composition analysis has shown a high presence of membrane phospholipids within the human retina, phosphatidylcholine being the primary form [40,41]. A recent comprehensive literature review from Longoni et al. thoroughly reviewed the pathways involved in lipid-mediated ROS production [30].

Furthermore, the role of mitochondrial DNA damage by ROS and an impairment in the autophagy pathway have been previously discussed as an essential element in the early elements of AMD pathogenesis [42,43]. Using a human in vitro model of AMD, it was shown that RPE cells underwent apoptosis, partially explained by their inability to upregulate the expression of SOD1 under oxidative stress [44]. Furthermore, the expression of peroxisome proliferator-activated receptor gamma coactivator-1α (PGC-1α)—a key regulator of mitochondrial biogenesis—was shown to be downregulated [44]. The second hallmark of AMD is geographic atrophy of the retina. The role of acute complement cascade activation in geographic atrophy development has been thoroughly reviewed in the past, especially the activation of C1q [45,46]. It was shown that subretinal macrophages are involved in the production of C1q [47]. Using a retinal ischemia/reperfusion (I/R) mice model, it was shown that retinal I/R upregulated C1q expression, activated microglia, and lead to retinal layer thinning [48]. Finally, basal linear and basal laminal deposits of drusen decompensate the RPE–Bruch’s membrane–choroid complex and subsequently lead to neovascularization [38]. Drusen deposition was shown to lead to hypoxia due to an impairment in oxygen and glucose transport at the choriocapillaris and subsequent vascular endothelial growth factor (VEGF) production within the retina [49]. VEGF activates Rac1 and NADPH in human choroidal endothelial cells, which in turn produces ROS [50]. It was recently shown that NADPH oxidases are activated by NOX4-p22^phox^, which is under the regulation of the transcription factor PU.1 [51]. These ROS then upregulate angiogenesis-promoting genes, thus contributing to AMD onset and progression.

### 3.2. The Role of Neuroinflammation

The abundance of lipids within the retina was shown to significantly contribute to the pro-oxidative and pro-inflammatory environment due to ROS-mediated lipid peroxidation processes [52]. Furthermore, a crucial role of oxidative stress in the pathogenesis of retinal diseases involves alterations within the immune system, which forms the backbone of neuroinflammation [53].

Dysregulations within the innate immune system are a major factor in AMD pathogenesis [54]. Using albino and pigmented mice strains, increasing age was shown to induce the subretinal accumulation of macrophages [55]. Furthermore, using transgenic mice without the expression of monocyte chemoattractant protein 1 (MCP-1), a team of researchers observed greater microglial activation within the subretina, as well as hypertrophy of RPE cells—a feature indicative of RPE cell death and retinal degeneration [56]. Macrophages are key mediators of neuroinflammation. The activation of macrophages can lead to the production of tumor necrosis factor (TNF) and interleukins (IL)-1, -6, -8, and -12 [57]. An in vitro experiment designed to induce chronic exposure to TNF-α in primary porcine RPE cells demonstrated the importance of TNF- α in neurodegenerative diseases such as AMD. Porcine RPE cells exposed to TNF- α exhibited hypertrophy, a decrease in gene expression involved in phototransduction, with an overall decrease in its immunomodulatory function [58]. Conversely, patients with geographic atrophy and AMD were shown to exhibit greater concentration levels of IL-6 and IL-8 [59]. A positive correlation between IL-6 concentration and geographic atrophy size was reported, as well as IL-8 concentration and neovascular AMD [59]. Furthermore, lL-1α, -1β, -4, -5, -10, -13, and -17 were also shown to be increased in the peripheral blood of patients with advanced AMD in comparison to healthy patients [60]. Expressions of these pro-inflammatory cytokines are known to be under the action of the nuclear factor kB (NFkB) [61] and mitogen-activated protein kinases (MAPK) [62]. Overall, these findings support the pivotal role of the innate immune system in neuroinflammation and AMD pathogenesis.

## 4. Antioxidant Strategies for the Prevention and Management of AMD

### 4.1. Dietary Nutrients and Supplements

#### 4.1.1. Vitamins C and E

Vitamins C and E are antioxidative vitamins present in the human lens. Vitamin C is a water-soluble antioxidant that endogenously protects cellular materials against ROS and free radicals [63]. Regarding dietary intake, it is found in many foods, such as citrus fruits, potatoes, and tomatoes [64]. Vitamin E is a lipid-soluble antioxidant that inhibits ROS production during fat oxidation and free radical reaction propagation [65]. It is present in vegetable oils, nuts, seeds, and green leafy vegetables, among other sources [65].

The preventive role of vitamins C and E, mainly found in AREDS2 supplementation in the prevention of late AMD progression, is well known [63,66,67]. A recent meta-analysis, encompassing mostly results from the AREDS study, provided further evidence that AREDS2—containing vitamins C and E—delays the progression of late AMD, as well as geographic atrophy [68]. Furthermore, in a case–control study, low vitamin C and E intake was shown to be associated with nAMD [69]. Outside of late-stage prevention, it is important to note that studies have indicated that supplementary vitamins C and E do not yield an appreciable preventative effect for AMD in its early stages, nor in the prevention of AMD incidence in healthy subjects [70,71,72,73,74]. For example, in a randomized placebo-controlled clinical trial, Taylor et al. (2002) found that consuming daily vitamin E supplements did not have a significant effect on the development of early AMD [72]. Overall, these results endorse the use of vitamins C and E in individuals with moderate-to-severe AMD, with the aim of preventing AMD progression to later stages of disease.

#### 4.1.2. Lutein and Zeaxanthin

Lutein and zeaxanthin are structural isomers which act as potent scavengers of singlet oxygens and other free radicals in the eye [75]. As such, they are appreciable attenuators of oxidative damage. In the context of AMD, it is established that oxidative damage initiated by UVB penetration into the retina can have a detrimental effect on human retinal pigment epithelial cells, leading to oxidative stress through the upregulation of ROS and dysregulation of endogenous antioxidants, thus linking UVB to AMD [76,77]. The macula, which contains the highest pigment concentration, is the area that is most resistant to degeneration. Lutein and zeaxanthin are the only dietary carotenoids in the macula, and together, their peak absorption spectra allow them to filter out ultraviolet and blue light, mitigating this effect of light penetration [78]. They undergo oxidation and transformations to protect the macula [79], and thus a lack of these carotenoids could worsen AMD progression. Epidemiologically, lower macular pigment concentrations are a known risk factor for AMD, which is likely credited to the lack of lutein and zeaxanthin [75,80]. In addition, pre-treatment of ARPE-19 (a human retinal pigment epithelial cell type) cells with lutein or zeaxanthin has been demonstrated to prompt a significant reduction in UVB-induced damage and cellular ROS levels [81]. This observational and experimental evidence, in addition to their chemical and biological properties, suggests a role for lutein and zeaxanthin in AMD prevention.

Dietary sources for lutein include chicken egg yolk, basil, parsley, spinach, and other leafy green vegetables, as well as leeks, peas, and green peppers [82]. Zeaxanthin can be found in red pepper, corn tortillas/chips, and chicken egg yolk [82]. Epidemiological research has shown that the consumption of dietary lutein and zeaxanthin has an inverse correlation with nAMD [83,84]. Serum carotenoid levels also exhibit a significant negative association with AMD risk [85].

Clinical evidence points to better visual outcomes arising from lutein and zeaxanthin consumption. Hammond et al. (2014) conducted a double-blind, placebo-controlled study on visual outcomes from lutein and zeaxanthin supplementation [86]. They found that daily supplementation of the two over a year caused significant increases in macular pigment optical density (MPOD) and serum lutein and zeaxanthin levels compared to the placebo treatment. Chromatic contrast and photo stress recovery also showed significant improvement. Similarly, Loughman et al. (2021) conducted an 18-month, double-masked, randomized, placebo-controlled clinical trial testing the effects of lutein, zeaxanthin, and meso-zeaxanthin supplementation in participants with open-angle glaucoma [87]. They found that supplementation with these three carotenoids caused significant increases in MPOD and contrast sensitivity under glare conditions. In secondary analyses of a multicenter, double-masked clinical trial, AREDS2 researchers found that the daily consumption of supplements had a significant beneficial effect of lutein and zeaxanthin, in the lowest intake quintile, in attenuating progression to late AMD when compared to the no lutein/zeaxanthin group [88]. Other human studies have demonstrated similar effects of lutein and zeaxanthin supplementation such as promoting significant improvements in lutein serum concentration and various measures of visual performance [89,90].

However, there have been conflicting reports about the appreciability of lutein and zeaxanthin consumption on visual and anatomical outcomes. In a randomized parallel study of postmenopausal women, Olmedilla-Alonso et al. (2018) found that while lutein and zeaxanthin supplementation significantly increased participants’ serum levels of the two xanthophylls, there was no significant change in MPOD compared to the placebo [90]. This discrepancy in the significance of the carotenoids’ effect could be due to differences in formulation and dosing. [90] utilized doses of 6 mg lutein/2 mg zeaxanthin per day. Contrastingly, ref. [86,87] utilized larger daily doses of 10 mg lutein/2 mg zeaxanthin and 10 mg lutein/2 mg zeaxanthin/10 mg meso-zeaxanthin, respectively. Therefore, the lower dosage tested by Olmedilla-Alonso et al. could be responsible for the lack of substantial results. Overall, the current evidence points to a considerable preventative benefit of lutein and zeaxanthin supplementation for ocular health.

#### 4.1.3. The Age-Related Eye Disease Studies

The age-related eye disease studies (AREDS) 1 and 2 are landmark clinical trials that were designed to evaluate the impact of formulations with antioxidative properties on the clinical progression of AMD and cataracts. Given their prospective, double-blind, and randomized nature, they are considered to be amongst the most robust clinical trials aimed to investigate the use of antioxidants on disease progression. Since the early 2000s, multiple reports have been published.

The AREDS1 study’s first results came to light in the early 2000s [91]. The AREDS formulation, consisting of vitamin C, vitamin E, beta-carotene, copper, and zinc, was shown to reduce the risk of AMD progression by 34% over a median of 6.3 years in participants with a high risk of disease progression (i.e., heavy smokers and ethnicity) [91,92]. They estimated that if all patients with high-risk AMD would adhere to AREDS supplements, over 300,000 people (out of 8 million people) would avoid vision loss due to AMD progression within 5 years [93]. However, given the association between heavy smokers and the use of beta carotene on the incidence of lung cancer [94,95,96,97], that was also demonstrated in the AREDS study—2.1% of patients taking the AREDS formulation developed lung cancer, from which 91% were heavy smokers [88]—beta-carotone was substituted with lutein and zeaxanthin in AREDS2. Furthermore, the AREDS2 study sought to investigate the impact of omega-3 fatty acid supplementation (i.e., docosahexaenoic acid and eicosapentaenoic acid) on AMD progression [98]. The supplementation of omega-3 fatty acids alongside AREDS2 in individuals with intermediate AMD was shown to not have an impact on AMD progression. Although primary analysis of AREDS2 did not show a further decrease in AMD progression when compared to AREDS1 formulation, AREDS2 supplementation is favored given the risk of lung cancer associated with AREDS1 [99]. Furthermore, the clinical significance of AREDS was shown to be effective in patients with intermediate (stage 3) or advanced (stage 4) AMD [100]. The supplementation was shown to have a 25% risk reduction for disease progression over 5 years, in individuals with stage 3 and stage 4 AMD [91]. Furthermore, AREDS2 was shown to reduce the risk of moderate vision loss by 19% at 5 years [101]. Therefore, AREDS is beneficial for individuals with stage 3 and 4 AMD but offers no benefit for healthy individuals or those with early-stage (stage 1 or 2) AMD. Currently, the patent for AREDS2 is under the company Bausch + Lomb.

### 4.2. Pharmacological Anti-VEGF Treatments for Neovascular AMD

VEGF is an important regulator of ocular angiogenesis. VEGF levels in tears are elevated in patients experiencing age-related macular degeneration [102]. In clinical studies, analysis of the effect of VEGF antagonists on neovascularization has demonstrated their ability to suppress nAMD and improve visual function in nAMD patients [103,104]. Currently, anti-VEGF therapies are considered the gold standard for AMD treatment [105]. Drugs used to treat nAMD specifically include ranibizumab, brolucizumab, bevacizumab, aflibercept, and pegaptanib sodium (Figure 5).

#### 4.2.1. Ranibizumab

Ranibizumab is an anti-VEGF immunoglobin antibody fragment that binds all VEGF-A isoforms. Its recommended clinical dosage regimen is 0.5 mg every 4 weeks [106,107]. Although this is not as effective as maintaining a 4-week gap between treatments, less frequent injections of once every 3 months after four initial monthly doses can be conducted in conjunction with regular assessments [107]. Reports of its intraocular half-life vary, with measurements ranging from 7.19 to 9 days [106].

#### 4.2.2. Brolucizumab

Brolucizumab is a humanized single-chain anti-VEGF antibody fragment that binds all VEGF-A isoforms. Due to its smaller molecular mass compared to other treatment options, it has a more extended mode of action, is more soluble, allows for higher molar dosing, and more effectively penetrates ocular tissue [106,108]. It has a clinical dosing regimen of three 6 mg monthly intravitreal injections which then changes to one injection every 12 weeks [109]. As such, it is one of the more infrequent treatment options for AMD. Its intraocular half-life is 4.3 days [110].

#### 4.2.3. Bevacizumab

Bevacizumab is a full-length monoclonal anti-VEGF immunoglobin antibody binding all VEGF-A isoforms. While it was initially developed and approved for metastatic colorectal cancer treatment, it also has an off-label use in treating nAMD [106]. Due to its low cost and high availability, it is an accessible option for most nAMD patients [111]. Estimates of its intraocular half-life vary, ranging from 4.9 to 9.8 days in humans [112,113,114]. It is delivered via intravitreal injection at a dosage of 1.25 mg every 4 weeks [106].

#### 4.2.4. Faricimab

Faricimab is a bispecific antibody binding VEGF-A as well as angiopoietin-2 (Ang2).

Ang2 enhances angiogenesis occurring through VEGF [115], making it a target for inhibition in nAMD treatment. It is administered through intravitreal injection and has been recently approved for nAMD treatment in the USA in January 2022 [116]. It has an estimated mean apparent systemic half-life of 7.5 days [116]. Its normal dosage is 6 mg every 4 weeks for the initial four doses, followed by more spaced-out injections. The frequency of the injections can be extended up to 16 weeks based on visual acuity and optical coherence tomography evaluations, allowing for a more personalized treatment regimen [117]. Therefore, it is one of the most infrequent pharmacological injection options for AMD treatment.

#### 4.2.5. Aflibercept

Aflibercept is a recombinant fusion protein that forms VEGF traps targeting all isoforms of VEGF-A, VEGF-B, and placental growth factor, all of which are angiogenic [106]. It is delivered via a 2.0 mg intravitreal injection at an initial frequency of every 4 weeks for the first three injections, which then changes to every 8 weeks [109]. It has a half-life of 11 days in the eye [118]. In a study comparing the efficacy of 2 mg aflibercept and 0.05 mg bevacizumab delivered monthly via intravitreal injection for 3 months, aflibercept was found to exhibit a more prompt and long-lasting effect on AMD symptoms [119].

#### 4.2.6. Conbercept

Conbercept is a recombinant fusion protein composed of VEGF receptor 1 and 2 extracellular domains and the Fc region of human immunoglobin. It binds all VEGF isoforms and placental growth factor [120]. It has a demonstrated 4.2-day half-life in rabbits [121]. An effective dosing regimen for conbercept is three initial monthly intravitreal injections of 0.5 mg followed by injections every 3 months [122].

#### 4.2.7. Pegaptanib

Pegaptanib is an aptamer that acts as an inhibitor by binding VEGF_165_ specifically. It was the first anti-VEGF treatment approved for nAMD. Its dosing regimen is 0.3 mg every 6 weeks via intravitreal injection, and its intraocular half-life is 4 days [123,124]. Due to its specificity, it has a slower mechanism of action than other anti-VEGF treatments which are unselective [123]. It provides superior visual outcomes in patients exhibiting early AMD lesions [125].

### 4.3. Nanotechnology-Based Drug Delivery Systems for AMD Prevention and Treatment

Nanotechnology has been developed to enhance the delivery of common drugs used to treat nAMD and explore other anti-VEGF treatment options. The proceeding section will highlight some recently developed nanotechnologies formulated to improve AMD treatment dose and longevity (Table 1).

Highlighting some of these advancements, Mu et al. (2018) designed bevacizumab-loaded multivesicular liposomes to improve the intravitreal retention time of the drug. They found that in rabbit eyes, the bevacizumab-loaded liposomes had a stronger sustained release effect than bevacizumab alone (over 56 days) and were still able to effectively inhibit the thickness of laser-induced choroidal neovascularization (CNV) lesions [126].

In a similar vein, Gao et al. (2023) developed an injectable nanofiber hydrogel containing betamethasone phosphate (anti-inflammatory drug), CaCl_2_, and an anti-VEGF drug. Their formulation, delivered via intravitreal injection to mice with laser-induced CNV, increased the effective treatment time when compared to the anti-VEGF drug alone [127].

In another study, Zhong et al. (2024) conjugated anti-VEGF and anti-Ang2 aptamers to RNA nanoparticles, utilizing a mechanism of action similar in principle to that of faricimab [128]. They found that the subconjunctival injection of the RNA nanoparticles resulted in internalization of the particles by cells in the retina and retinal pigment epithelium. Among the tested configurations of the particles, many exhibited sizeable antiangiogenic effects, with the larger RNA square particles (SQR-VEGF-Ang2) exhibiting strong potential as an effective anti-VEGF treatment suitable for posterior eye delivery.

Alternatively, testing a non-invasive mode of delivery, Sun et al. (2024), engineered therapeutic protein eyedrops consisting of penetratin hyaluronic acid-liposomes loaded with conbercept [129]. Penetratin is a cell-penetrating peptide, which was used to enhance ocular penetration. Hyaluronic acid was used as a retina-targeting ligand. In conjunction, these two components of the liposome allow for non-invasive AMD therapy, as the liposomes penetrate the ocular barrier and target conbercept to the retina. They found that the peak intraocular concentrations of conbercept were 11.5 times higher with the administration of the liposomes as compared to conbercept alone. Administration of the liposome treatment had an equivalent effect to intravitreal conbercept injection in inhibiting CNV formation in mice with laser-induced CNV.

Testing a novel treatment option, Chen et al. (2024) examined a nanomedicine delivery system composed of RGD peptide-modified liposomes loaded with 2-deoxy-D-glucose (2DG) in an effort to target endothelial cell metabolism [130]. 2DG interferes with *N*-glycosylation, which was sufficient to inhibit VEGF receptor 2 downstream signaling, resulting in significant inhibition of laser-induced CNV and decreased CNV lesion size in mice. In addition, the RGD-modified liposome vehicle was able to improve cellular uptake in vascular endothelial cells both in vitro and in vivo without impacting the drug release profile. In conjunction, these results suggest a promising therapeutic role for these liposomes in the context of nAMD. In particular, there may be a strong benefit of this treatment for patients who have been unresponsive to current anti-VEGF treatments.

In addition to other nanotechnologies, nanozymes, which are nanoscale-sized particles exhibiting catalytic activity, have also demonstrated an antioxidative role in AMD treatment [133]. Cupini et al. (2023) demonstrated that the intravitreal injection of platinum nanoparticles increased photoreceptor survival and attenuated retinal inflammation in rats with light damage [131]. Additionally, Shin et al. (2022) developed a noninvasive cerium oxide (CeO_2_) delivery wafer, called the Cerawafer, to attenuate oxidative stress in the retina [132]. CeO_2_ is known to exhibit catalytic properties in oxidation–reduction reactions and has been shown in previous studies to reduce neuroglial inflammation and mitochondrial ROS levels [134]. Specifically, the cerium cycles between Ce^3+^ and Ce^4+^ oxidation states, allowing it to catalyze reactions with superoxide and hydrogen peroxide to effectively eliminate ROS from the cell [135]. Shin et al. found that the nanoparticles delivered via Cerawafer downregulated VEGF and scavenged ROS in the retina, underscoring another possible mode of treatment in patients unresponsive to anti-VEGF drugs [132].

### 4.4. Gene Therapy for Neovascular Age-Related Macular Degeneration Treatment

Clinical studies have shown that pharmaceutical nAMD treatments in the form of VEGF antagonists yield strong improvements in visual acuity. However, in reality, clinical practice shows less substantial results [136,137,138]. This is because injections in clinical practice are more infrequent than in controlled clinical studies due to undertreatment or being less often maintained [136,137,138]. Nanotechnology-based AMD treatment has been tested in animal models to improve the longevity of these drugs. However, current studies on nanotechnology-based AMD treatment, while promising, have been largely experimental. Clinical trials testing gene silencing treatments targeting VEGF-A with small interfering RNA have also been conducted but have not progressed past phase 3. This is due to challenges such as RNA instability, non-specific targeting, limited bioavailability, and the need for frequent treatment, hindering its successful application [105]. As such, gene therapy in the form of ocular gene transfer may be a better option for the long-term treatment of nAMD, as it requires less frequent treatment and has shown a strong potential for success in early-phase clinical trials. The proceeding section will provide an overview of results from current gene therapy treatments in the clinical testing stage for nAMD treatment (Table 2).

#### 4.4.1. Ixoberogene Soroparvovec

Ixoberogene soroparvovec (ixo-vec, also called ADVM-022) is an adeno-associated virus vector encoding aflibercept, which, as previously discussed, is a widely utilized anti-VEGF drug for nAMD treatment. In a phase 1, open-label, prospective two-year clinical study, Khanani et al. (2024) tested the safety and efficacy of ixo-vec [139]. They provided a single dose of ixo-vec via intravitreal administration in two different doses: 2 × 10^11^ and 6 × 10^11^ vector genomes (vg) per eye. They found that vision and central subfield thickness were stable two years post ixo-vec injection. They also noted 80% and 98% reductions in the low- and high-dose groups, respectively, in annualized supplemental injections of aflibercept delivered at the investigators’ discretion (to maintain best corrected visual acuity and avoid disease progression). In terms of the evaluation of ixo-vec safety, they found no systemic adverse events in participants. They noted two serious ocular treatment-emergent adverse effects that were likely related to ixo-vec: asymmetric progression of preexisting dry AMD and recurrent uveitis, after corticosteroid therapy for inflammation was discontinued. They also found mild-to-moderate ocular treatment-emergent adverse events that were dose-dependent, the most common of which was anterior chamber inflammation in 7/15 of the low-dose group and 11/15 of the high-dose group participants and vitreal cell inflammation in 3/15 of the low-dose group and 8/15 of the high-dose group participants. However, this intraocular inflammation could be managed by topical corticosteroids, with no anterior chamber or vitreous inflammation noted in low-dose participants by the end of the study.

In conjunction, these results indicate the clinical benefit of ixo-vec for nAMD management, as it is well tolerated at low doses, provides less frequent need for injection, and is able to maintain vision and improve anatomical outcomes compared to aflibercept injection alone.

#### 4.4.2. ABBV-RGX-314

ABBV-RGX-314 is an anti-VEGF-A antigen-binding fragment expressed by an adeno-associated virus serotype 8 vector, that is currently being explored as a treatment for nAMD. In preclinical studies, it has been shown to suppress nAMD symptoms in mice in a dose-dependent fashion [146]. Campochiaro et al. (2024) conducted a phase 1/2a dose-escalation clinical study of ABBV-RGX-314 to assess its safety and tolerability [140]. They found that the dose at which the subretinal injection of the vector was well tolerated was 3.0 × 10^9^ to 1.6 × 10^11^ genome copies per eye. Doses of 6 × 10^10^ genome copies and higher caused sustained RGX-314 protein levels for at least 2 years in the aqueous humor and best corrected visual acuity that was consistent with or improved from before treatment. Of the 42 participants, 13 experienced serious adverse effects over the course of the study, but only one event was possibly related to RGX-314. It was found in a participant who was administered a large dose of 2.5 × 10^11^ genome copies per eye. They did not observe any clinically recognizable immune responses in participants. In conjunction, these results render ABBV-RGX-314 a strong candidate for future use in clinical practice. The current research thus provides a guideline for the safe use of ABBV-RGX-314 in nAMD treatment. However, it is limited by its novelty, as later phase clinical trials are required to confirm this treatment’s efficacy and safety long term.

#### 4.4.3. Angiostatin and Endostatin

Angiostatin and endostatin are angiogenesis-inhibiting proteins that counter VEGF to help terminate neovascularization endogenously [147]. In mouse models, increased expression of angiostatin and endostatin has caused the suppression of CNV [148,149]. Campochiaro et al. (2017) conducted an open-label, phase 1, dose escalation study of RetinoStat^®^, which is an equine infectious anemia virus (EIAV)-based lentiviral vector co-expressing angiostatin and endostatin [141]. They administered the vector via a subretinal injection to participants with advanced nAMD in the following doses: 2.4 × 10^4^, 2.4 × 10^5^, and 8.0 × 10^5^ transduction units. They found that the doses were well tolerated with little ocular inflammation and that aqueous humor levels of angiostatin/endostatin increased in a dose-related manner. Of the 21 participants, 8 exhibited sustained expression of angiostatin/endostatin for 2.5 years, and 2 exhibited sustained expression for over 4 years. Mean levels of angiostatin/endostatin peaked 12–24 weeks post-injection. They had one serious procedure-related adverse effect, where the procedure caused a macular hole in one participant, which was later resolved.

Overall, angiostatin and endostatin’s role as angiogenesis inhibitors and preclinical research demonstrate their potential for nAMD treatment. Campochiaro and colleagues provide a guideline for the safe use of this treatment and present lentiviral vectors as a safe platform for ocular gene therapy. However, current research is limited by a lack of advanced clinical studies exploring angiostatin/endostatin and its direct effect on nAMD treatment and progression, which thus becomes an avenue for future exploration.

#### 4.4.4. FTL-1

FTL-1 is a gene for a receptor that inhibits VEGF-A and prevents angiogenesis. It has a demonstrated ability to significantly inhibit CNV in primate and rat models [150]. Constable et al. (2016) conducted a phase 2a randomized controlled clinical trial studying rAAV.sFLT-1, which is a recombinant adeno-associated vector expressing sFLT-1 [142]. This was following a phase 1 clinical trial where they found that rAAV.sFLT-1 was well tolerated in patients 50 years and older with nAMD [151]. Participants received subretinal injections with pars plana vitrectomy in two groups: one group received ranibizumab injections, and the other received a subretinal injection of rAAV.sFLT-1 at a dose of 1 × 10^11^ vg (which was determined in their phase 1 study) and ranibizumab retreatment injections as needed. They noted no serious ocular adverse events and no systemic adverse events. However, they did note 51 mild ocular adverse events in the gene therapy group (which had 21 participants), with 26 of these being attributed to the study procedure. Of these, 2 were likely related to rAAV.sFLT-1: eye inflammation and anterior chamber inflammation, both of which were resolved. They found that rAAV.sFLT-1 injection resulted in the median number of ranibizumab retreatments being reduced to 2.0, as compared to the control group value of 4.0.

Another clinical study has also been conducted testing sFLT-1 gene transfer as a treatment for nAMD. Heier et al. (2017) conducted a phase 1, open-label, clinical trial of the intravitreous injection of AAV2-sFLT01, which is an AAV2 vector expressing sFLT-1 [143]. In terms of the dose-escalation portion of the study, they found no dose-limiting toxic effects and thus used their highest tested dose, 2 × 10^10^ vg, as the maximum tolerated dose. Two out of ten patients in the maximum dose group experienced adverse events: pyrexia and intraocular inflammation (which was resolved with topical steroid prescription). In lower-dose cohorts, zero participants exhibited detectable aqueous humor concentrations of sFLT01 protein 52 weeks post-injection. Of the patients who received the maximum dose, 5 out of 10 had protein concentrations above the limit of quantification (mean 53.2 ng/mL). Overall, [143] showed promising transgene expression in some patients and demonstrated their treatment’s safety and tolerability for a range of doses. It would be beneficial for future studies to explore larger tolerable doses of AAV2-sFLT01 during treatment, to maximize post-treatment sFLT01 protein levels. Both Heier et al. and Constable et al. provide strong guidelines for the safe dosage of the studied gene therapy products. However, due to the early phase of these studies, more research on the efficacy of the treatments against nAMD can further consolidate the validity of this treatment.

#### 4.4.5. Pigment Epithelium-Derived Factor

Pigment epithelium-derived factor (PEDF), which is an endogenous anti-angiogenesis factor, was tested in the first clinical trial for gene therapy of nAMD [144,152]. In a randomized, phase 1 clinical trial, Campochiaro et al. (2006) gave 28 patients an intravitreous injection of AdPEDF.11, which is a viral vector encoding human PEDF [144]. They found that seven of the participants displayed mild intraocular inflammation and six had increased intraocular pressure (which was resolved with topical medication). They found that the results were dose-related, with cohorts receiving the larger, 10^8^ to 10^9.5^ particle unit dose exhibiting no increases in median nAMD lesion area along with significantly decreased neovascularization after 12 months. Cohorts with lower doses, on the other hand, had an increase in median lesion size. This research provides insight into the longevity of AdPEDF.11 injections and defines a tolerable dose of the treatment that achieves the attenuation of nAMD symptoms. However, it is limited by its early phase, as further clinical research has not been conducted to assess its long-term efficacy.

#### 4.4.6. KH631

KH631 encodes a human VEGF receptor fusion protein in a recombinant adeno-associated virus 8 vector. Ke et al. (2023) developed this product and conducted a preclinical evaluation of its efficacy in non-human primates [145]. They found that a single dose of 3 × 10^8^ vg per eye delivered via subretinal injection was able to induce sustained transgene expression in the retina for a duration of over 96 weeks. Additionally, it was able to attenuate the progression and formation of grade IV CNV lesions. KH631 is currently in the early stages of phase 1 and 2 clinical testing [153].

## 5. Challenges and Limitations of Antioxidants and Novel Therapeutic Approaches

While current treatments for AMD are effective and of a high caliber, there do exist challenges for each mode of treatment. With regard to antioxidant dietary supplements such as vitamins C and E, their ability to attenuate the late-stage progression of AMD is well established. However, there is a lack of a consensus reflecting the preventative effects of vitamins C and E for early AMD in particular [70,71,72,73,74]. In a similar vein, while lutein and zeaxanthin have an established role in the improvement in visual and anatomical outcomes in patients with ocular disease, their efficacy as a treatment for AMD is limited by a lack of clinical research investigating the effect lutein and zeaxanthin alone on AMD pathogenesis. Overall, there is not enough information to support the use of antioxidants in the prevention of early-stage AMD, and thus this is an important direction for future research.

As aforementioned, anti-VEGF drugs exhibit strong efficacy and are the current gold standard for nAMD treatment. However, clinical practice does not demonstrate as strong of a positive effect on patients with nAMD due to lower adherence to injection schedules and undertreatment [136,137,138]. Additionally, intravitreal injection, which is the most common mode of delivery for anti-VEGF drugs, can result in ocular adverse events such as increased intraocular pressure, endophthalmitis, and conjunctival hemorrhage [122]. While nanotechnology-based drug delivery methods do work to improve the delivery of these drugs, current research is largely preclinical. This lack of clinical research forms a barrier to the understanding of the safety and toxicity of these nanotechnology-based treatments when translating these advancements into clinical applications.

Gene therapy also helps to improve the longevity and dosage of AMD treatments and is largely focused on anti-VEGF treatments specifically. However, it is important to consider the implications of VEGF inhibition in the eye. For example, reports that the continuous dosing of anti-VEGF drugs results in a higher incidence of macular atrophy raises concern and exemplifies the importance of the critical assessment of current standards of treatment [154,155]. In addition, treatment with anti-VEGF drugs has been demonstrated to induce epithelial-mesenchymal transition in retinal pigment epithelium [156]. This can be counteracted with an anti-fibrotic protein (CCN5), which may be considered in the implementation of anti-VEGF treatments [156].

## 6. Recommendations and Future Directions

Given the aging population globally, the prevalence and incidence of AMD is facing a shift in its trend. As studies underlying the role of oxidative stress in ocular pathologies are growing at a spectacular rate, the possibility to use exogenous antioxidants for the prevention and treatment of these diseases has gained interest. In the light of our literature review, no recommendations can be given for the use of vitamins C and E, lutein, and zeaxanthin for the prevention of cataracts and AMD as independent agents; the results are currently conflicting and further studies are necessary. However, the recommendations suggest the use of AREDS2 for the prevention of AMD stage 3 and 4 progression. Therefore, the evidence-based suggestions for clinicians are to recommend the use of AREDS2 formulation in patients with AMD to prevent disease progression. Major challenges exist in the delivery methods of antioxidant agents to the posterior eye segment; the BRB limits the bioavailability of drugs within the human eye [157]. Leveraging these challenges is the next step in the management of ocular pathologies. Further studies to optimize antioxidant delivery to the posterior eye segment are required. Nevertheless, a safe approach to delay or prevent AMD progression or incidence is to act on risk factors: AMD patients are advised to stop smoking. However, although the association with the other risk factors and AMD is controversial, patients should be advised to be cautious with their diet and alcohol usage as well.

## 7. Conclusions

The human eye is a complex organ, composed of several structural elements across the anterior and posterior segments, and it faces daily environmental and endogenous stress. AMD is amongst the leading causes of vision impairment in the posterior eye segment. Exogenous and endogenous stressors are known to induce ROS production, which is the backbone of disease initiation and progression. Numerous studies have sought to investigate the potential of antioxidant treatments for the management of AMD. However, given the physical challenges involved in drug delivery to the eye compartments, further studies are required to establish the efficacity of antioxidants in AMD prevention in clinical trials.

## Figures and Tables

**Figure 1 biomedicines-12-01579-f001:**
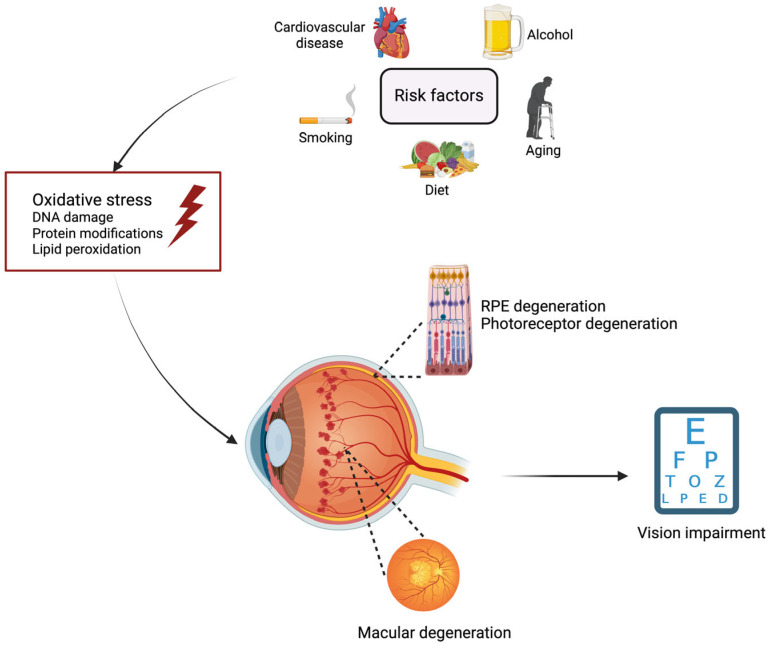
Schematic illustration of the pivotal role of oxidative stress in age-related macular degeneration: from risk factors to disease progression. Numerous risk factors have shown a differential association with the development or progression of age-related macular degeneration (AMD). Aging is the most well-defined risk factor of AMD. The association of AMD progression with alcohol, smoking, diet, and the presence of cardiovascular diseases remains, however, controversial. The pro-oxidative environment leads to DNA damage, lipid peroxidation, and protein modifications, which are the mediators of retinal changes (e.g., macular degeneration, retinal pigment epithelium (RPE) degeneration, photoreceptor degeneration, and neovascularization (represented with the blood vessels within the eye)). Altogether, these consequences can lead to vision impairment. Image created with BioRender.com (accessed on 22 June 2024).

**Figure 2 biomedicines-12-01579-f002:**
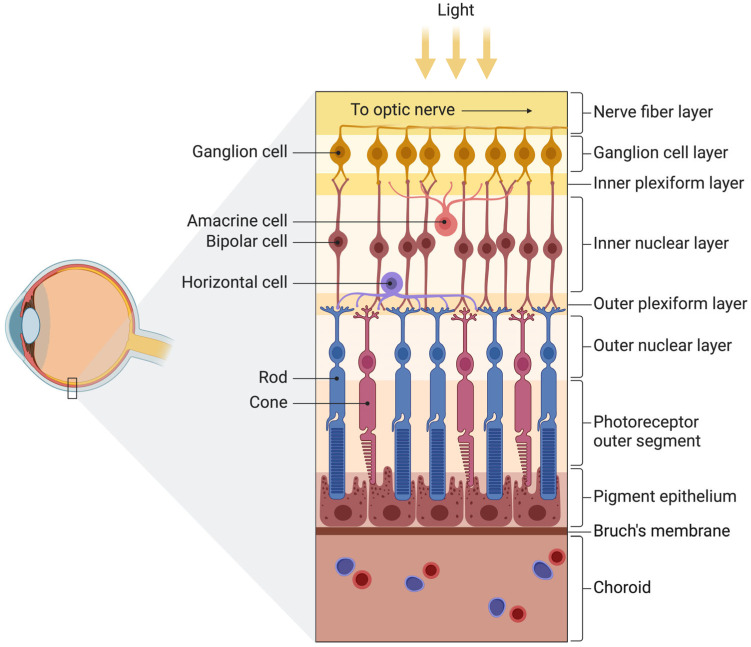
Schematic illustration of the retinal structure. The human retina consists of 10 distinctive layers, each encompassing crucial cells involved in phototransduction. Alterations to the retinal structure form the backbone of retinal diseases. Reprinted from “Structure of the Retina”, by Biorender.com (2024). Retrieved from https://app.biorender.com/biorender-templates and created from BioRender.com (accessed on 22 June 2024).

**Figure 3 biomedicines-12-01579-f003:**
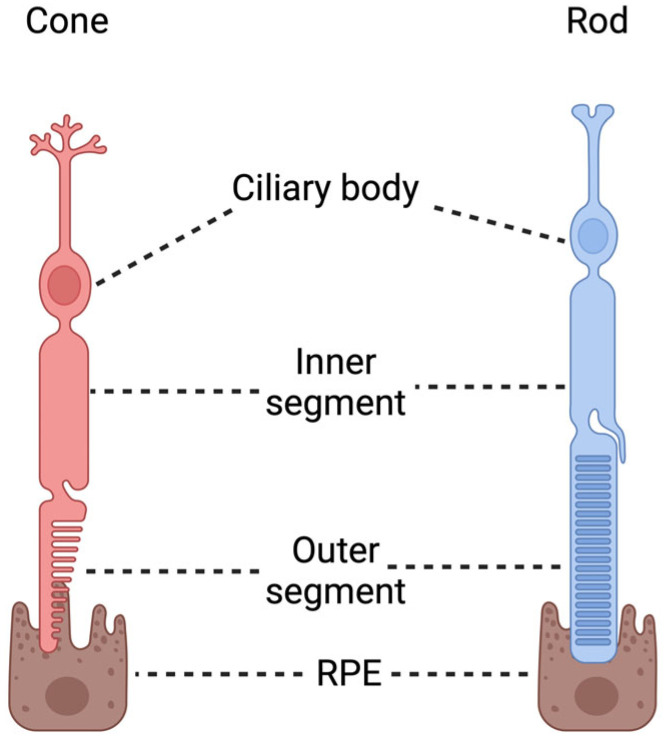
Schematic illustration of photoreceptor structure. Human photoreceptors consist of cones and rods. Their structure can be divided into three distinct compartments: the ciliary body, the inner segment, and the outer segment. Abbreviation: RPE, retinal pigment epithelium. Image created with BioRender.com (accessed on 22 June 2024).

**Figure 4 biomedicines-12-01579-f004:**
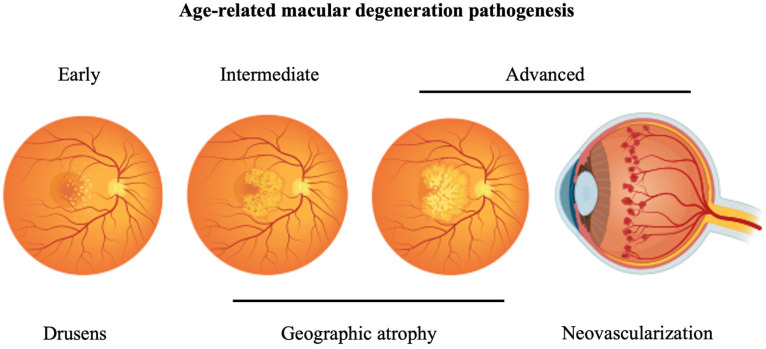
Schematic illustration of age-related macular degeneration pathogenesis. Age-related macular degeneration (AMD) involves drusen deposits and/or geographic atrophy in its early stages. As the disease progresses, the geographic atrophy enlarges. Wet AMD, also known as neovascular AMD (nAMD), is characterized by angiogenesis of retinal blood vessels due to vascular endothelial growth factor (VEGF) secretion by endothelial cells. The figure was created with BioRender.com (accessed on 22 June 2024).

**Figure 5 biomedicines-12-01579-f005:**
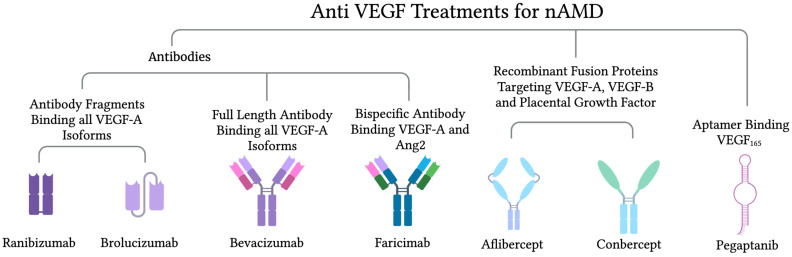
Current anti-VEGF treatments for neovascular age-related macular degeneration. The figure was created with BioRender.com (accessed on 22 June 2024).

**Table 1 biomedicines-12-01579-t001:** Summary of most recent nanotechnologies formulated to improve treatment outcomes for age-related macular degeneration.

Drug	Nanotechnology Used	Outcomes	Reference
Bevacizumab	Bevacizumab encapsulated with multivesicular liposomes	Exhibited stronger sustained release than bevacizumab alone, attenuated the thickness of laser-induced CNV lesions in rabbits.	[126]
Betamethasone phosphate, anti-VEGF	Nanofiber hydrogel containing CaCl_2_ and anti-VEGF drug delivered via intravitreal injection	Increased effective treatment time of anti-VEGF when compared to using anti-VEGF alone.	[127]
Anti-VEGF and anti-Ang2 aptamers	RNA nanoparticles	Particles were internalized in retina and retinal pigment epithelium cells. Particles exhibited anti-VEGF effects and promising posterior eye delivery.	[128]
Conbercept	Eye drops formulated as penetratin hyaluronic acid-liposomes loaded with conbercept	Allowed for non-invasive penetration of the ocular barrier and targeting of product to the retina, caused an 11.5-fold increase in peak intraocular conbercept concentration compared to conbercept alone, and matched the effect of the intravitreal injection of conbercept on the inhibition of laser-induced CNV.	[129]
2-deoxy-D-glucose	RGD peptide-modified liposomes	Inhibited VEGFR-2 signaling, attenuated laser-induced CNV, and decreased CNV lesion size in mice. RDG-modified liposomes improved cellular uptake.	[130]
Platinum nanoparticles		Decreased retinal inflammation and enhanced photoreceptor survival in rats with light damage.	[131]
CeO_2_	Cerawafer	Decreased VEGF expression and scavenged retinal ROS.	[132]

**Table 2 biomedicines-12-01579-t002:** Summary of current novel gene therapies for the treatment of neovascular age-related macular degeneration.

Intervention and Route	Company	Transgene Product	Vector	Outcomes	Clinical Trial Phase and Reference
Ixo-vec, intravitreal	Adverum Biotechnologies	Aflibercept	AAV2	Doses of 2 × 10^11^ and 6 × 10^11^ vg/eye caused 80 and 90% decreases, respectively, in annualized supplemental injections of aflibercept. Intervention was well tolerated at low doses.	[139]
ABBV-RGX-314, subretinal	RegenxBio AbbVie	RGX-314	AAV8	Determined that the intervention was well tolerated at doses of 3.0 × 10^9^ to 1.6 × 10^11^ genome copies per eye, with doses above 6 × 10^10^ genome copies generating sustained RGX-314 expression for over 2 years post-injection.	[140]
Retinostat, subretinal	Oxford BioMedica plc	Endostatin and angiostatin	EIAV	Intervention was well tolerated at doses of 2.4 × 10^4^, 2.4 × 10^5^, and 8.0 × 10^5^ transduction units. Endostatin/angiostatin levels in the aqueous humor were dose-dependent and peaked 12–24 weeks post-injection. Some participants exhibited long-term sustained expression of endostatin/angiostatin.	[141]
rAAV.sFLT-1, subretinal	Adverum Biotechnologies	sFLT-1	AAV2	Intervention dose of 1 × 10^11^ vg reduced the median amount of ranibizumab retreatment following vitrectomy from 4 to 2. No serious ocular adverse events from intervention.	[142]
AAV2-sFLT01, intravitreous	Genzyme	sFLT-1	AAV2	Intervention was well tolerated and a maximum tolerated dose was not determined as all doses tested were tolerated. 5/10 participants that received the highest tested dose of 2 × 10^10^ vg exhibited sFLT01 protein concentrations above the limit of quantification.	[143]
AdPEDF.11, intravitreous	GenVec	Pigment-derived epithelial factor	Adenoviral vector	Intervention doses of 10^8^ to 10^9.5^ particle units halted increases in median nAMD lesion area and mitigated neovascularization after 1 year.	[144]
KH631, subretinal	Chengdu Origen Biotechnology Vanotech	Anti-VEGF fusion protein	AAV8	Intervention delivered at 3 × 10^8^ vg per eye causes sustained expression for more than 96 weeks in non-human primates. Intervention prevents the progression and formation of grade IV CNV lesions in non-human primates.	[145]

## Data Availability

No new data were created or analyzed in this study. Data Sharing is not applicable to this article.

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
