# Peer review of "The Role of Reactive Oxygen Species in Age-Related Macular Degeneration: A Comprehensive Review of Antioxidant Therapies"

_biomedicines, 2024, doi:10.3390/biomedicines12071579_

Round 1

Reviewer 1 Report

Comments and Suggestions for Authors

The Review “Comprehensive review “The Role of Reactive Oxygen Species in Cataract Formation and Age-Related Macular Degeneration: A Comprehensive Review” focuses on examination of the roles of reactive oxygen species (ROS) in the pathogenesis of cataracts and age-related macular degeneration (AMD), making it a highly informative resource. The review also covers, analysis of the oxidative stress mechanisms involved in these ocular conditions, which enhances the insight of the causing pathological processes. There are three figures and three tables representing the review content for better understanding. Overal, the review may act as a good educational resource for healthcare students and professionals in ophthalmology, offering deep insights into the potential of antioxidants in mitigating the burden of cataracts and AMD. However, it reqiurs minor revision as suggested before publication.

My obervationa and comments to improve the manuscripts are as follows:

1.      Abstract: Line 16: The article concludes by synthesizing current…This sentence need to rewrite with better clarity.

2.      Line 27: …research findings, clinical trial data, and meta-analyses to provide evidence-based recommend…. Did author perform meta analysis?

3.      Line 51: USD ppp, predefine abbreviations.

4.      Line 54: (6–10), please cite at specific line or sentences. Avoid bulk citation as its difficult to coreleated which citation is for what content.

5.      There should be figure on etiology ROS. Refere: Fig. 1 of https://www.mdpi.com/2076-3921/12/7/1379 and there are many similar work present in literature.

6.      Figure 1 and 2 are not useful as its imple explaining the well known facts. I suggest to remove and  illustrate more useful content in manuscript.

7.      Inclusion of clinical finding related to antioxidants and novel therapeutic approaches.

8.      Patents releated to products ourcome of novel therapeutic approaches .

9.      Please include challnages and limiatation of antioxidants and novel therapeutic approaches. 

Author Response

Dear Reviewer,

Thank you very much for your thorough review and valuable suggestions. We greatly appreciate your feedback and have made the necessary revisions to address each of your comments. Below are our responses and the corresponding changes made to the manuscript:

Abstract: Line 16: The sentence has been rewritten for better clarity.

Line 27: We apologize for any confusion. We did not perform a meta-analysis. The sentence has been revised to accurately reflect the content: "...research findings and clinical trial data to provide evidence-based recommendations."

Line 51: The abbreviation "USD ppp" has been defined in the text for clarity.

Line 54: The bulk citation has been addressed by specifying citations at the relevant lines or sentences. This ensures that each reference is clearly associated with the corresponding content.

Figure on etiology ROS: We have included a new figure that illustrates the etiology of reactive oxygen species (Figure 1).

Figures 1 and 2: Based on your suggestion, Figures 1 and 2 have been removed. We have incorporated more useful content and illustrations that add value to the manuscript.

Inclusion of clinical findings: We have included additional clinical findings related to antioxidants and novel therapeutic approaches to enhance the manuscript's comprehensiveness.

Patents related to products: We have included a few lines (lines 347-348) discussing a patent related to the outcomes of novel therapeutic approaches, highlighting recent advancements and innovations. While there are numerous patent numbers available online, we believe that including an exhaustive list may not be relevant for the focus of this manuscript. Therefore, we have opted to provide a concise version instead.

Challenges and limitations: A new section discussing the challenges and limitations of antioxidants and novel therapeutic approaches has been included (section 5), providing a balanced perspective on the topic.

We are confident that these revisions have improved the manuscript and addressed your concerns. Thank you once again for your constructive feedback and for helping us enhance the quality of our work.

Sincerely,

Reviewer 2 Report

Comments and Suggestions for Authors

The present review is aimed at providing evidence of the antioxidant efficacy of a variety of intervention to cope with cataract and AMD both pathologies are strongly related to oxidative stress. The authors start with the physiology of the lens and retina but while the physiology of the lens is reasonably described there are several criticisms about the retina starting from the complete lack of information about the functional organization of retinal neurons (see lines 178-179), paper quotes lot of literature but, in my opinion, lack to mention the appropriate ones related to the many ways to arrive to oxidative stress, starting from the relevant energy requirement of photoreceptors (see for example Longoni, B. & Demontis, G.C. Polyunsaturated Lipids in the Light-Exposed and Prooxidant Retinal Environment. Antioxidants 2023 ) to get an idea. Subsequently they treat lens and AMD in parallel I think that authors should separate the two topics if nothing else the role of neuroinflammation in the progression of the disease induced by oxidative stress has to be discussed in depth specifically in the retina. The organization of the entire review deserves to be critically evaluated to offer a clear analysis of the present and future available treatments. The paper is difficult to read because it appears mainly a long list of suggestions without a clear link although the amount of the information is relevant and it might deserve a better critical discussion.

Author Response

Dear Reviewer,

We would like to extend our sincerest appreciation for your thorough and insightful review of our manuscript. We are grateful for your valuable comments, which have significantly contributed to enhancing the quality and clarity of our work. We have carefully considered each of your points and made the necessary revisions to address them comprehensively. Below, we provide a detailed response to your comments:

  1. Lack of information about the functional organization of retinal neurons (lines 178-179): Thank you for pointing out this oversight. We have added a detailed section on the functional organization of retinal neurons, ensuring that the physiological aspects of the retina are as comprehensively covered as those of the lens. This section now includes a description of the various types of retinal neurons and their roles in visual processing.
  2. Literature on the many ways to arrive at oxidative stress: We appreciate your suggestion to include more relevant literature regarding oxidative stress. We have incorporated additional references, including the suggested article by Longoni and Demontis (2023), to provide a more thorough overview of the pathways leading to oxidative stress, particularly focusing on the energy requirements of photoreceptors and the prooxidant environment in the retina.
  3. Separation of lens and AMD topics: In response to your recommendation, we have opted to remove all details on the lens and focus solely on AMD. After separating the two topics into distinct sections, we found that the manuscript became excessively long, which is not suitable for a review article. By focusing solely on AMD, we are now in a better position to provide a detailed and thorough discussion on AMD as per your request.
  4. Role of neuroinflammation in AMD progression: We have expanded the discussion on the role of neuroinflammation in AMD progression induced by oxidative stress. This section now offers a comprehensive analysis of the mechanisms involved and the current understanding of how neuroinflammation contributes to the disease process, supported by recent research findings.
  5. Organization and critical evaluation of the review: We have undertaken a critical re-evaluation of the entire review’s organization to enhance readability and coherence. The manuscript has been restructured to provide a clearer analysis of the present and future available treatments, with each section logically flowing into the next. We have also included more critical discussions to link the various pieces of information, ensuring that the review is not merely a list of suggestions but a cohesive and comprehensive narrative.
  6. Improved readability and critical discussion: To address your concern about readability, we have revised the manuscript to ensure that each section is clearly linked, providing a smooth transition between topics. We have also enhanced the discussions throughout the manuscript, being more critical, as well as offering a more balanced and thorough evaluation of the evidence presented.

We hope that these revisions meet your expectations and significantly improve the quality of our manuscript. We are committed to delivering a high-standard review and appreciate your constructive feedback, which has been instrumental in this process.

Thank you once again for your valuable comments and suggestions.

Warm regards,